



# Dust Radiative Effects on Atmospheric Thermodynamics and Tropical Cyclogenesis over the Atlantic Ocean Using WRF/Chem Coupled with an AOD Data Assimilation System

Dan Chen[1], Zhiquan Liu[1], Chris Davis[1], Yu Gu[2]

[1]National Center for Atmospheric Research, Boulder, Colorado, USA

[2]University of California, Los Angeles, Los Angeles, California, USA

*Correspondence to:* Zhiquan Liu (liuz@ucar.edu) and Dan Chen (dchen@ucar.edu)

## Abstract

This study investigated the dust radiative effects on atmospheric thermodynamics and tropical cyclogenesis
over the Atlantic Ocean using WRF-Chem coupled with an aerosol data assimilation (DA) system. MODIS
AOD data were assimilated with the Gridpoint Statistical Interpolation three-dimensional variational DA
scheme to depict the Saharan dust outbreak events in 2006 summer. Comparisons with Ozone Monitoring
Instrument (OMI), AErosol RObotic NETwork (AERONET) and Cloud-Aerosol Lidar and Infrared Pathfinder
Satellite Observation (CALIPSO) observations showed that the system was capable of reproducing the dust
distribution. Two sets of 180-hr forecasts were conducted with the dust radiative effects activated (RE_ON) and
inactivated (RE_OFF), respectively. Differences between the RE_ON and RE_OFF forecasts showed that low-
altitude (high-altitude) dust inhibits (favors) convection owing to changes in convective inhibition.  Heating in
dust layers immediately above the boundary layer increases inhibition whereas sufficiently elevated heating
allows cooling above the boundary layer that reduces convective inhibition. Semi-direct effects are also noted in
which clouds are altered by thermodynamic changes, which then alter cloud-radiative temperature changes.
The analysis of a tropical cyclone (TC) suppression case on Sep. 5 shows evidence of enhanced convective
inhibition by direct heating in dust, but also suggests that the low-predictability dynamics of moist convection
reduces the determinism of the effects of dust on time scales of TC development (days).



## 1. Introduction

Mineral dust, one of the most abundant aerosol species in the atmosphere, has important weather and climatic effects through its influence on solar and terrestrial radiation and the radiative and physical properties of clouds (e.g., Grassl, 1975; Sokolik et al., 1998; Quijano et al., 2000; Ginoux et al., 2001; Ramanathan et al., 2001; Lau et al., 2009; Zhao et al., 2010). The Sahara desert over North Africa is the largest source of mineral dust in the world. During the summer months, the Saharan dust outbreaks are associated with a dry and hot well-mixed layer (Saharan Air Layer, SAL) extending to ~500 hPa over the North Atlantic Ocean. Saharan dust, often propagating downstream along the SAL to the Atlantic Ocean, can modify the SAL and its environment by changing the energy budget (e.g., Su et al., 2008; Chen et al., 2010; Zhao et al., 2010) through either scattering and absorption of sunlight (direct effect, e.g. Carlson and Benjamin, 1980; Zhu et al., 2007; Rosenfeld et al., 2008; Wong et al., 2009; Chen et al., 2010); thermodynamic effect on clouds (semi-direct, e.g. Hansen et al., 1997); and altering cloud microphysical processes (indirect effects, e.g. Kaufman and Koren, 2006). As pollution and smoke aerosols can increase or decrease the cloud cover, this duality in the effects of aerosols forms one of the largest uncertainties in climate research (Koren et al., 2005, 2008; Kaufman et al., 2006). The term 'semi-direct effect' was originally introduced by Hansen et al. (1997) to describe the impact of absorbing aerosols that cause cloud evaporation and thus dissipate the cloud prematurely. Modeling (Lohmann and Feichter 2001; Cook and Highwood 2004) and observational studies (Ackerman et al., 2000) showed that when aerosols are embedded within clouds, increased shortwave absorption could reduce relative humidity and subsequently change cloud cover, e.g., decrease cloud cover especially at the upper level. However, the semi-direct effect of dust is sensitive to the position of the dust layer relative to clouds (Choobari et al., 2014). If the dust layer is located below clouds, heating within the dust layer can enhance convection and thus cloud cover. Absorbing aerosol above the cloud top may inhibit the vertical development of clouds and enhance the horizontal development by the suppression of entrainment due to the increase in temperature above the cloud (Johnson et al., 2004; Koch and Del Genio, 2010). All these lead to either a positive or negative radiative forcing, as in semi-direct effect (Gu et al., 2010, 2015), which is poorly understood for Saharan dust.

Tropical cyclones (TCs) in the Atlantic basin often develop from Meso-scale Convective Systems (MCSs) embedded within African easterly waves (AEWs) that originate over West Africa (Landsea, 1993). Whether easterly waves spawn tropical cyclones depends on the maintenance of a mesoscale region of rotation with deep moisture and a minimum of convective inhibition. Dust-induced thermodynamic changes, distinguished from indirect cloud effects, may modulate the environment of convection, thereby altering its frequency and other properties. Such direct and semi-direct effects can, in principle, alter the behavior of tropical disturbances. Several observational and numerical studies (e.g., Karyampudi and Carlson, 1988; Karyampudi and Pierce, 2002; Dunion and Velden, 2004; Evan et al., 2006; Jones et al., 2007; Wu, 2007; Sun et al., 2008, 2009; Pratt and Evans, 2009; Reale et al., 2009, 2014; Shu and Wu, 2009; Chen et al., 2010) have shown evidence for aerosol-induced intensification or weakening of TC development. For example, by using satellite data, Dunion and Velden (2004) have found that the dry SAL can suppress Atlantic TC activity by increasing the vertical wind shear and stabilizing the environment at low atmospheric levels. They also



suggested that convectively driven downdrafts caused by the SAL dry air can be an important inhibiting factor for TCs. Based on a single case of Sep. 5-12, 2006, Sun et al. (2009) also found that with the temperature and humidity assimilation of AIRS observations during dust outbreak periods, the dry and warm SAL was better simulated such that the development of tropical disturbances was inhibited along the southern edge of the SAL due to enhanced downdrafts. By conducting a composite study of 274 cases using AIRS relative humidity, Shu and Wu (2009) provided evidence that the SAL can affect tropical cyclone intensity in both favorable (in the initial development) and unfavorable (subsequent development) manners. Bretl et al. (2015) confirmed the complexity of dust radiative effects by analyzing differences in hurricane genesis and frequency with radiatively active and inactive dust in the aerosol-climate model ECHAM6-HAMAll. These studies indicated that the Saharan dust could indeed influence Atlantic TC genesis while the relationship and mechanisms are not fully understood. Most of recent studies (e.g. Jenkins et al., 2008; Zhang et al., 2009; Hazra et al., 2013; Wang et al., 2014) aimed to reveal the coupled results of microphysical and radiative effects or to emphasize the microphysical impacts of dust on TCs – seeding hurricanes with dust acting as CCN (Cloud Condensation Nuclei). Fewer papers are on solely radiative effects and there is still no consensus whether the radiative effects of dust inhibit or favor TC development.

The magnitude of the radiative impact of dust highly depends upon the distribution of the dust concentration and its optical characteristics. While the modeling of the spatial distribution of the Saharan dust outbreak and its optical depth remains uncertain and challenging, AOD data assimilation (DA), combining satellite-derived AOD observations with numerical model output, has proved to be skillful at improving aerosol and AOD forecasts (Collins et al., 2001; Liu et al., 2011). Liu et al. (2011, hereafter L11) implemented AOD DA within the National Centers for Environmental Prediction (NCEP) Gridpoint Statistical Interpolation (GSI) three-dimensional variational (3DVAR) DA system coupled to the Goddard Chemistry Aerosol Radiation and Transport (GOCART) (Chin et al., 2000, 2002) aerosol scheme within the Weather Research and Forecasting/Chemistry (WRF/Chem) model (Grell et al., 2005). Verification results demonstrated improved aerosol forecasts from AOD DA over a week-long period in a study of a dust storm event over East Asia. Reale et al. (2014) used the NASA GEOS-5 global data assimilation and forecast system to investigate the aerosol direct radiative effects and their impact on the atmospheric dynamics. The improved aerosol and meteorological analysis fields by assimilating MODIS AOD, satellite, and conventional data sets were used to initialize the forecast by GEOS-5. They did two sets of simulations (with dust radiative active and inactive) for the period of Aug. 15- Sep. 17, 2006. The differences between the two runs revealed the dust suppression effect on Atlantic tropical cyclone – strong dust effects make the environment less conducive to tropical cyclone development.

In this study, we will further extend our previous air quality-oriented study (Chen et al., 2014; Schwartz et al., 2014) to investigate the dust-radiation effects on thermodynamics and TC genesis in the Atlantic Ocean using the GSI-WRF/Chem coupled AOD DA system. Model description and experimental design are presented in section 2. In section 3, we investigated the dust radiative effects on atmospheric thermodynamics emphasizing temperature and Convective Inhibition Energy (CIN) changes caused by dust at different altitudes. Then we examined the hypothesis that the dust at different altitudes induced different temperature and CIN





changes above the top of boundary layer, which in turn may modulate the frequency and strength of the TC genesis in either a favorable or un-favorable manner. Statistics based on long-term periods and case analysis were both provided. Conclusions are given in Section 4.

## 2. Model Description and Experimental Design

Version 3.4.1 of WRF/Chem was employed with "online" coupled meteorological and chemical processes. The model domain with 36-km horizontal grid spacing covers the Atlantic Ocean and the Sahara desert (see Fig. 2). There are 57 vertical levels extending from the surface to 10 hPa. Aerosol direct and semi-direct effects are allowed in WRF/Chem by linking the optical properties of GOCART aerosols (OC and BC with 2 size bins, sulfate, dust with 5 size bins and sea salt with 4 size bins) to the Goddard Space Flight Center Shortwave radiation scheme (Chou and Suarez, 1994). The scattering/absorption coefficients and single-scattering albedos are calculated by the "aerosol chemical to aerosol optical properties" module built in WRF/Chem (Barnard et al., 2010). The dust refractive indexes are wavelength dependent. An average single scattering albedo of ~0.9 at 600 nm is reported by Zhao et al. (2010). Aerosol radiative effects on longwave radiation and aerosol indirect effects (i.e., aerosols as Cloud Condensation Nuclei or Ice Nuclei in microphysics scheme) were not implemented for GOCART with the WRF/Chem version used. The WRF single-moment 6-class microphysics scheme and the Grell-Devenyi ensemble cumulus parameterization (Grell and Devenyi, 2002) were used. The dust emission flux is computed as a function of probability source function and surface wind speed (Ginoux et al., 2001). Similar to dust uplifting, sea salt emissions from the ocean are highly dependent on the surface wind speed and calculated as a function of wind speed at 10 m and sea salt particle radius (Chin et al., 2002). The lateral boundary conditions (LBCs) for meteorological fields were provided by the NCEP GFS forecasts. LBCs for chemistry/aerosol fields were idealized profiles embedded within the WRF/Chem model.

NCEP's GSI 3DVAR DA system was used to assimilate the total 550-nm AOD (hereafter "AOD") retrievals (Collection 5) from MODIS sensors onboard Terra and Aqua satellites, as described in L11. Retrievals of Deep Blue products over desert (Hsu et al., 2004) and dark targeting products over ocean and vegetated land (Remer et al., 2005) were assimilated in this study. Only AOD retrievals marked with the best quality flag were assimilated (see L11 for more detail). Several analyses of the collection 5 MODIS over-ocean AOD product show that MODIS AOD data accuracies are statistically comparable with other standard aerosol satellite products and can be used in operational aerosol assimilations (Yang et al., 2011; Ahn et al., 2014). L11 implemented AOD DA by using the Collection 5 MODIS 550-nm AOD data in studying a dust storm in East Asia and verification results demonstrated improved aerosol forecasts from AOD DA over a week-long period. More results from our DA simulations, comparing with OMI and AERONET, are given in section 3.

The GSI 3DVAR system calculates a best-fit "analysis" considering the observations (AOD in our case) and background fields (a short-term WRF/Chem forecast in our case) weighted by their error characteristics. The Community Radiative Transfer Model (CRTM) (Han et al., 2006; Liu and Weng, 2006) is used as the AOD observation operator in GSI to transform the GOCART aerosol profiles into AOD. Each aerosol species in 3-D





is used as analysis variable of 3DVAR, and the 3-D mass concentrations of the aerosol species are analyzed in a one-step minimization procedure constrained by the observation and background error covariances (BECs). The AOD observation error is assumed to be spatially uncorrelated and modeled in the same way as that in L11. To more accurately reflect the dust-dominated feature, the background error covariance statistics for each aerosol

variable were calculated by utilizing the "NMC method" (Parrish and Derber, 1992) based upon the one-month WRF/Chem forecasts for the dust outbreak month of August 2005. Standard deviations and horizontal/vertical correlation length scales of the background errors (separated for each aerosol species) were calculated using the method described by Wu et al. (2002). Note that only column-total 550-nm AOD was assimilated and no aerosol speciation information was contained in the observations. It is important to have the phenomena-

specific background error statistics to allow for an appropriate adjustment of individual species. As a function of vertical model level, the domain-averaged standard deviations of the background errors for each GOCART aerosol species are shown in Fig. 1. Consistent with the dust outbreak period, the standard deviations of dust errors are one or two orders of magnitude larger than those of other species. A larger background error of dust allowed larger adjustment of dust field, which is crucial for the aerosol analyses of the Sahara dust outbreak

period in this study. Further details for the AOD DA system, including the algorithm, the observation operator, and the modeling of the background and observation error covariances can be found in L11.

We selected the 2006 summer which has been investigated for similar interests in several studies as aforementioned (Sun et al., 2009; Real et al., 2014). The MODIS AOD assimilation experiment was conducted using this GSI-WRF/Chem coupled system. The experiment initialized a 6-hr WRF/Chem forecast every 6-hr

starting from 00 UTC 1 Jul. to 00 UTC 18 Sep., 2006. In this period, several TC genesis cases occurred along with the outbreaks of the Saharan dust. The first week of simulation is taken as spin-up time and the results from a later period (Jul. 8 - Sep.18) were analyzed. Meteorological initial condition (IC) for each forecast came from the GFS analysis (i.e., no regional meteorological DA). GSI 3DVAR updated GOCART aerosol variables by assimilating MODIS AOD at 12 UTC and 18 UTC (when AOD observations were available) and using the

previous cycle's 6-hr forecast as the background. Two sets of 180-hr forecasts were made at 00 UTC of each day (Jul. 8 - Sep. 18) from this 6 hourly cycling procedure. The only difference between the two sets of forecasts is that the aerosol radiative feedback is activated (RE_ON) or not (RE_OFF). Note that meteorological ICs in the two 180-hr forecasts are the same (from the GFS analysis) and the forecast differences of meteorological fields between the RE_ON and RE_OFF experiments result solely from the differences in

aerosol radiative effects.

It should be noted that in this study, the microphysical influence of dust aerosols acting as cloud condensation nuclei or ice nuclei is not included in the model set up. Only the direct and semi-direct radiative effects are considered in our study. As the cloud cover calculations in the model are relevant to the RH/T changes, the cloud-induced semi-direct effect is indeed represented as a consequence of the direct effect in the

model. Because the model set up is not able to simulate microphysical effects related to dust, our focus is on thermodynamic changes that mainly influence area-mean profiles of temperature and moisture that define the environment for moist convection. Thus, the relationship to cyclogenesis centers on the mechanism by which



convection is favored or suppressed by the dust-radiative interaction over a region that includes a nascent cyclone, not on detailed cloud processes within that region.

## 3. Results

### 3.1 Verification of assimilated dust outbreaks

Figure 2 shows the 2006 Jul., Aug., and Sep. period-averaged 550-nm AOD in the assimilation experiment compared with Level-3 OMI/Aura 483.5-nm AOD in the study domain. Model simulated results are from the 6-hr forecast valid at 18 UTC which is closest to the OMI pass time. As the OMI data only provide the 342.5, 388.0, 442.0, 463.0 and 483.5 nm AOD, no direct comparison to 550-nm AOD can be made. High 550-nm AOD (0.4-0.7) over the Atlantic Ocean is shown in the DA experiment. Although the simulated 550-nm AOD values are still lower than the OMI 483.5-nm AOD, the locations and shapes of the dust tongues are well represented. The black rectangles define the main dust region and also the Main Cyclogenesis Region (MCR). To distinguish the effects near to and far from the dust source region, the MCR is divided into two regions: the eastern MCR (east of 30 °W) and the western MCR (west of 30 °W).

The simulated 550-nm AODs at three AERONET sites along the dust transport path are also shown in Fig. 3 to verify the aerosol DA performance. The locations of the three AERONET sites are shown as blue dots in Fig. 2 (left panel). The observations obtained from AERONET are interpolated to 550-nm for comparisons (Eck et al., 1999). At Capo Verde and Dakar sites near the dust source region, high AOD values during the dust outbreak periods were successfully reproduced. At La Parguera near the end of the dust path, simulation also captured the AOD fluctuations well.

It is also important to verify the vertical distributions of AOD in the model, as the semi-direct effect of dust is sensitive to the position of the dust layer with respect to clouds. Figure 4 shows the vertical distributions of modeled 550-nm AOD compared to AOD retrievals from the CALIOP instrument on board the Cloud-Aerosol Lidar and Infrared Pathfinder Satellite Observations (CALIPSO) satellite (Winker et al., 2009). The CALIOP AOD product is provided at 5 km (60 m) horizontal (vertical) resolution and was averaged to match the model resolution (36 km) and vertical grid before comparing to the model output. As the CALIPSO path is rather narrow and data availability is limited, it is difficult to find matching dust outbreak cases without any missing data. Here three cross-sections along CALIPSO paths on Jul. 29 and Sep. 5 are shown. The outbreaks of Saharan dust and the westward transport are shown on both days (left panels). In addition, the top-left panel on Jul. 29 shows a maximum of AOD in the western Atlantic which is caused by an earlier dust outbreak on Jul. 23 (not shown). Although CALIPSO shows more high AOD values below 500 m over the ocean, the overall patterns of the model simulated AOD are similar to CALIPSO observations with high AOD values around 4 km near the source region (top panel) which extended to lower levels (below 2 km) over the ocean.

### 3.2 Dust-induced vertical thermodynamic structure changes


Given reliable dust fields and associated optical properties with the assimilation of MODIS AOD, the forecast differences between RE_ON and RE_OFF allow us to have a better understanding of the dust-radiative effect and its feedback onto the thermodynamics and dynamics. The analysis in this section is based on the hourly output from the first 96 hours of all forecasts from Jul. 8 - Sep. 18. To distinguish the effects near and far from the African coast, the analysis is conducted for the eastern MCR (near-desert region) and the western MCR (far-from-desert region) separately (Fig. 2). To understand the effects of the vertical distribution of dust, the averaged AOD vertical profiles for both the eastern MCR and western MCR are given in the Contoured Frequency Altitude Diagrams (CFADs) in Fig. 5. The frequencies in the eastern MCR show that the dust from the Sahara Desert is mostly elevated during the westward transport process and the AOD peaks at 700 hPa with values around 0.02-0.03. There are also few cases of deep layers of dust that are not elevated to 700 hPa altitude but with relatively high AOD below 850 hPa. For the western MCR, the frequencies of deep layers of dust are higher than that in the eastern MCR indicating the dust depositing process along the transport path. We define the cases of "Deep Layer of Dust" as AOD larger than 0.015 at 900 or 950 hPa but smaller than 0.02 at 700 hPa. The total sampling ratios are 8.5% and 10.7% in the eastern and western MCR respectively. As the remaining cases show the common characteristic of elevated dust, we also categorized them as the second type - "Elevated Dust". For the four types of dust ("Elevated Dust"/"Deep-Layer Dust" in the eastern/western MCR), thermodynamic/dynamic changes in the first 96 hours between RE_ON and RE_OFF were averaged hourly for the Jul. 8 - Sep. 18 forecasts. The changes (RE_ON – RE_OFF) in temperature (T), relative humidity (RH), vertical velocity (W), Convective Inhibition (CIN), and Buoyancy for boundary layer parcels are shown as pressure-forecast time series in Figs. 6-9(a). To investigate how the clouds were changed due to dust-induced heating effects and also to separate the changes by short- and long-wave cloud radiative effects, the mixing ratios of clouds (liquid plus ice water) and the differences of temperature tendencies (K/day) due to long wave and short wave radiative effects are also given in Fig. 6-9(b-c).

For the "Elevated Dust" in the eastern MCR (Fig. 6), the dust was more concentrated above the top of boundary layer (Fig. 6a). Significant positive T anomalies occur around 600-800 hPa due to the dust radiative absorption and the warming has a diurnal variation with largest signal during the strongest solar heating period (12-18 UTC). Obvious negative T anomalies occur both above and below the dust. The negative T anomalies above the dust lag the positive anomalies within the dust and also have a diurnal variation. Below the dust, negative T anomalies start to occur from the night of the second day and do not have a significant diurnal cycle above 950 hPa; but the anomalies extend to lower atmosphere during the morning (07-12 UTC). Correspondingly, positive RH anomalies also occur below dust layer above 950 hPa and the changes show as early as in the night of the first day – several hours after the strong radiative heating during the first afternoon. The positive RH anomalies have the similar pattern as that of the T anomalies, in terms of occurrence time and vertical structure after the second day. As the clouds fields (middle panels) show no significant low-cloud changes and heating rates changes due to LW (bottom panels) at 800-950 hPa layer, the cooling below 800 hPa may result from a combination of reduced radiation below the dust layer and adiabatic upward motion driven by the heating above, similar to the "Elevated Heat Pump" mechanism (Lau et al., 2006, 2009; Gu et al. 2015). It is notable that the T and RH anomalies above the dust layer strengthen with time although the elevated AOD





weakens; it might be related to high ice clouds and heating rate changes above 600 hPa. The cooling and moistening at 850-950 hPa, above the top of boundary layer, lead to significant reduction of CIN during the nighttime. It should be noted that CIN is calculated for parcels lifted from each pressure level, and Fig. 6a shows the value of CIN plotted at the parcel origination level. Buoyancy differences for boundary layer parcels

more clearly correspond to differences of temperature above the boundary layer (Fig. 6a). In the cases that boundary-layer parcels are not strongly negatively buoyant, decreased CIN/increased buoyancy just above the top of boundary layer may weaken the inversion and lead to the destabilization of the lower troposphere with respect to deep convection.

For the "Elevated Dust" in the western MCR (Fig. 7), the warming due to dust is stronger than that in

the eastern MCR but no cooling occurs below the dust layer, although moistening is still present (top panels). The differences of the temperature anomalies above the boundary layer in the two regions may come from the different clouds and heating rates changes (Fig. 6b,c and 7b,c). In both the western and eastern MCR, the low clouds are right at the top of boundary layer. In the western MCR, the cloud amounts are larger than that in the eastern MCR (as the air becomes moister from eastern to western MCR) and the cloud changes are also more

prominent. As the dust is concentrated immediately above the low clouds, it is possible that the cloud concentrations are decreased at 900-950 hPa due to the evaporation effect. As the longwave radiation of dust is not included in the model, the heating rates changes due to LW are mainly from the changes of clouds. As cloud-top cooling and cloud-bottom heating normally occur associated with longwave (LW) radiative effects (Fu and Liou 1993), the vertical changes of clouds cause significant positive heating rates anomalies at 900-950

20   hPa for the western MCR (Fig. 7c) that offset the adiabatic cooling there. Unlike the cooling near surface in the eastern MCR, slight warming occurs in the western MCR. It is noticeable that the vertical velocity also starts to show negative anomalies at the top of boundary layer from the second day that become prominent on the third day. Weakened upward motion, or strengthened downward motion, is consistent with the CIN increase there.

For the "Deep Layer of Dust" in the eastern MCR (Fig. 8), the dust layers are thicker than that of

"Elevated Dust" and extend into the boundary layer. The dust absorption causes large positive temperature anomalies below 800 hPa starting from the first day. But the temperature changes become more complicated from the third day when the clouds changes are more prominent. As the dust not only covers the low-cloud layer but also extends to the altitudes above the cloud layer, low clouds change in a mixed manner and redistribute vertically - decreasing at the top of the boundary layer and increasing at the altitudes above. The

changes in low level clouds lead to an increase in heating rate around 900 hPa and a decrease below 900 hPa. In addition to the changes of low liquid clouds at low levels, ice clouds at high levels also change (mostly increase) significantly. While outgoing longwave is mostly controlled by high clouds, it might be the reason for the other maximum differences of heating rates around 200-400 hPa. However, those factors still do not fully explain the negative temperature anomalies starting from the third day, indicating that some other complicated

processes occurred in the case of deep layers of dust in the eastern MCR during the later portions of forecasts which are not revealed in our analysis here. For this reason, only the results of the first two days for this type of dust are taken into account in our hypothesis later.



The changes in cases of "Deep Layer of Dust" in the western MCR (Fig. 9) are more explainable. In addition to the dust-absorption-induced heating above the boundary layer, the vertical changes of low-level clouds also lead to large positive heating rates changes at 900 hPa which enhance the positive temperature anomalies and CIN increase there. Ice clouds both increase and decrease during different times which leads to a less coherent signal at high levels. Similar to the case of "Elevated Dust" in the western MCR (Fig. 7), vertical velocity starts to decrease at the top of the boundary layer from the second day and this becomes prominent from the third day onward. The statistics here (Fig. 6-9) show how dust induces changes of temperature, and also cloud, in the lower troposphere that modulate convective inhibition.

### 3.3 Synthesis

### 3.3.1 Effect on Deep Convection

Based on the above statistics in section 3.2, we hypothesize that in the early stage of forecasts the altitude of the dust determines whether there is warming or cooling just above the top of the boundary layer. Convection is sensitive to temperature changes just above the boundary layer because this is the layer in which convective inhibition is typically found. For this region, CIN is also very sensitive to the 700 hPa temperature. In the cases where significant heating occurs just above the boundary layer (deep layer of dust in the eastern and western MCR) or at the 700-800 hPa layer (elevated dust in the western MCR), at an elevation where there might be already some convective inhibition, the heating makes the inhibition stronger. This suppresses convection and prevents the TC development. With elevated dust, the heating occurs more near the middle troposphere. Below the dust layer, the atmosphere may experience cooling because of adiabatic ascent forced by the heating above, or by reduced solar absorption near the top of the boundary layer. In regions where air is relatively dry and no significant cloud-induced semi-direct effect occurs to weaken the cooling (the eastern MCR), this cooling could remove inhibition, although convection may not develop at all if it is too dry. In regions with greater relative humidity above the boundary layer (the western MCR), the cooling might be obscured due to the cloud-induced semi-direct effect.

According to our hypothesis, along with temperature and CIN changes elevated dust-induced TC intensification in the eastern MCR and suppression in both the eastern and western MCR would occur under certain conditions, especially when the atmosphere is in neutrally stable condition that little perturbation would change the direction. For the elevated dust-induced TC intensification in the eastern MCR, however, it requires strict conditions as the two key factors are contradictory: no significant clouds-induced heating requires that the air is relatively dry while TC development requires enough moist for convection. For the suppression cases, the possibilities in the western MCR might be larger than that in the eastern MCR, since the vertical velocity decrease significantly (Fig. 7 and 9) in the western MCR that works in the same direction of the CIN increase at the top of the boundary layer. Besides, we need to keep in mind that instability to moist convection is necessary but not sufficient for the occurrence of organized convective complexes. CIN changes by the temperature anomalies at the top of the boundary layer might be the earliest signal but not the only factor that leads to the intensification/suppression occurrence.





### 3.3.2 TC case

To interpret the dust-radiative effects on TC development, we choose a case where there is a clear difference in behavior between forecasts with the dust radiative feedback operating, and the feedback turned off. We view the effects of dust on the thermodynamics and the occurrence of deep, moist convection within the vortex. Microphysical effects of dust within the convection itself are not included. Here, the Sep. 5 forecast is used to illustrate a case of suppressed development when dust-radiative forcing is active. There was no TC in the real atmosphere on this date in the location examined here. The vortex developed within a region of cyclonic shear vorticity to the east of hurricane Florence.

Figure 10 clearly shows the suppression of the vortex development that occurred between 72 and 90 hours (Sept. 8). TC suppression case occurred during the 72-90 hour forecasts initialized from 00UTC 5 Sep. The surface wind was almost the same at 00UTC 8 Sep in RE_ON and RE_OFF but it weakened significantly in RE_ON starting from 06UTC and the differences were most prominent at 18UTC. This case is also the same one as in Sun et al. (2009). The cyclone centers were tracked by GFDL Tracker. Figure 11 shows the cyclone-area-averaged (defined as red rectangles in Fig. 10) AOD, and differences (RE_ON – RE_OFF) for T, RH, W, CIN and buoyancy (for boundary layer parcels) as a function of height and forecast valid time. The time series start from 00UTC 5 Sep. Figure 11 clearly shows the heating due to dust at 800-900 hPa and also the CIN increase/buoyancy decrease above the PBL (black dots) in the first 72-hour forecast, ahead of the suppression.

Figure 12 depicts the west-east cross sections across the center of MCS (red dot in Fig. 10) at 06UTC 8 Sep when different surface wind pattern started to show in RE_ON. Figures 12a-b are the (u, $100 \times w$) streamlines and temperatures from RE_ON and RE_OFF respectively. Figures 12c-d are the differences of temperatures and RH between the two forecasts (RE_ON-RE_OFF). Figure 12c shows different circulation patterns with weakened upward motion in RE_ON around the center of MCS at ~44.6 °W (Fig. 10). Consistent with Sun et al. (2009), the lack of upward motion weakened the large-scale boundary layer convergence which drives the deep and moist convection. Over time, these processes resulted in the erosion of the MCS. The above phenomena are very consistent with our hypothesis and also the statistic about the deep layer of dust in the western MCR: heating caused CIN increase is the earliest signal and weaken upward motion/strengthen downward motion are the accompany results.

Figure 13 show the clouds and heating rates changes for the cyclone region. The largest AOD due to dust outbreak occurred at 600-950 hPa for the first two days (Fig. 11). The low clouds were mostly at 850-950 hPa, thus the low clouds were mostly embedded in the dust layers for the first two days. As the long-wave radiation of dust is not hooked up in the model, it can be seen that in the very beginning (during the daytime of Sep. 5) dust-radiation interaction was mostly through the direct effect of shortwave absorption (Fig. 13e) around the dust maximum that reduced the amount of low clouds slightly (Fig. 13c). The changes of cloud distribution (also including the high clouds) were associated with the long-wave radiation changes consequently which caused obvious clouds changes starting from Sep. 7. When cloud decreases, less heating at the cloud base





(cooling effect) and less cooling at the cloud top (warming effect) should be expected. Here the differences of LW heating rates (Fig. 13d) illustrate this. Although clouds started to change significantly from Sep. 7, It is still difficult to interpret the relationships with other factors (e.g. convection, vertical motion) that are associated with limited predictability.

**4.  Conclusions**

In this study, the WRF-Chem model coupled with an AOD DA system was used to investigate the dust radiative effects on atmospheric thermodynamics and tropical cyclogenesis over the Atlantic Ocean during the summer period of 2006 (Jul. 8- Sep. 18). 6-hourly analysis/forecast cycling experiment with the assimilation of MODIS 550-nm AOD was conducted from 00UTC 1 Jul to 00UTC 18 Sep. Comparisons with OMI show that
the system depicts the dust outbreak and transport with acceptable accuracy. The dust outbreak and transport features are also consistent with the AERONET and CALIPSO observations in terms of AOD. 180-hr forecasts were then initialized from 00UTC of each day (Jul. 8 – Sep. 18) with the same meteorological ICs interpolated from the GFS analysis but with aerosol radiative effects active (RE_ON) and inactive (RE_OFF). Statistics of dust-induced thermodynamic changes were conducted based on the differences of the first 96 hour forecasts of
the two simulations. Four types of dust - "Elevated Dust"/"Deep Layer of Dust" in the eastern MCR/western MCR were categorized and investigated to understand the effects of dust vertical distribution and the distance to dust source region.

Statistical analysis shows that the altitude of the dust caused different thermodynamic changes especially for temperature at the top of the boundary layer, which resulted from the combination of three
different processes: 1) direct radiative heating of dust within the dust layer; 2) adiabatic ascent (and cooling) beneath a sufficiently elevated dust layer; (3) changes in low clouds (and associated radiative effects) that result from dust-induced thermodynamic changes.  The last process is sensitive to the position of the dust layer relative to clouds and also to the cloud amount. The combinations of these processes caused different temperature changes for different cases at the top of boundary layer: a) significant cooling for the cases of
elevated dust in the eastern MCR (near-desert region where air is dry) by combined processes 1 and 2, b) slightly warming for elevated dust in the western MCR by the three processes (where cloud-induced heating offsets adiabatic cooling), and c) significant warming for deep layers of dust in both regions by processes 1 and 3. The statistics showed the temperature anomalies occur as early as in the first 1-2 days, then become prominent during the 2-3 days, and remain consistent to the end of the 96-hour forecast; the only exception is
for the deep layer of dust in the eastern MCR where the later forecasts (after 48 hours) showed much more complicated thermodynamic responses that result from changes in deep convection.

As CIN is sensitive to the temperature just above the boundary layer, we assume that dust-induced temperature changes may modulate convection frequency and intensity.  For suppression cases, heating near or above the top of boundary layer leads to a CIN increase and buoyancy decrease for boundary layer parcels,
thereby suppressing convection.  However, it should be noted that heating within the boundary layer itself, due to dust, can increase CAPE by increasing parcel temperatures and buoyancy. This would offset the warming



near and above the top of the PBL. The possibility for suppression to occur in the western MCR is larger than that in the eastern MCR, as the statistics show decreased vertical velocities in the western MCR that contribute to an increase of convective inhibition. While destabilization due to elevated dust is possible in the eastern MCR, it requires more strict conditions that the moist in the air is in a range that does not cause too much clouds and heating rates changes but meanwhile provide enough energy for convection development- which might be contradictory in reality.

The Sep. 5 TC suppression case proved our hypothesis on the deep layer of dust. The case study also revealed that weakened upward (strengthened downward) motions and clouds-induced semi-direct effects were also found at the same time when the suppression occurred. It should be noted that although the simulations are for the two-and-half months (Jul. 8- Sep. 18, 2006), the samples of the TC suppression are limited. We only selected one suppression case which might not be typical enough. More investigations are needed to better understand these complex interactions so as to further evaluate our hypothesis.

**Acknowledgements**

This work is partially supported by grants from U.S. Air Force Weather Agency. The authors thank Junmei Ban for help with the AERONET graphs and Hailing Zhang for help with the GFDL Tracker package. NCAR is sponsored by the National Science Foundation.

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







**Figure 1.** Domain-averaged standard deviations of background errors (μg/kg) as a function of height for each
aerosol variable.



**Figure 2.** Time-averaged 550-nm AOD in data assimilation experiment (left) compared with Level-3 OMI/Aura
483.5-nm AOD (right). Averaged period: Jul. 8-31 (top), Aug. 1-31 (middle), Sep. 1-17 (bottom). The
rectangles are defined as the Main Cyclogenesis Region (MCR), including the eastern MCR and the western
MCR. The blue dots (left panels) are the AERONET sites used in Fig. 3. The scales on the left and right color
bars are different.



**Figure 3.** Model-simulated 550-nm AOD (black) and AERONET 550-nm AOD (red) at three sites. The locations of the three sites are shown as blue dots in Fig. 2 (left panel).



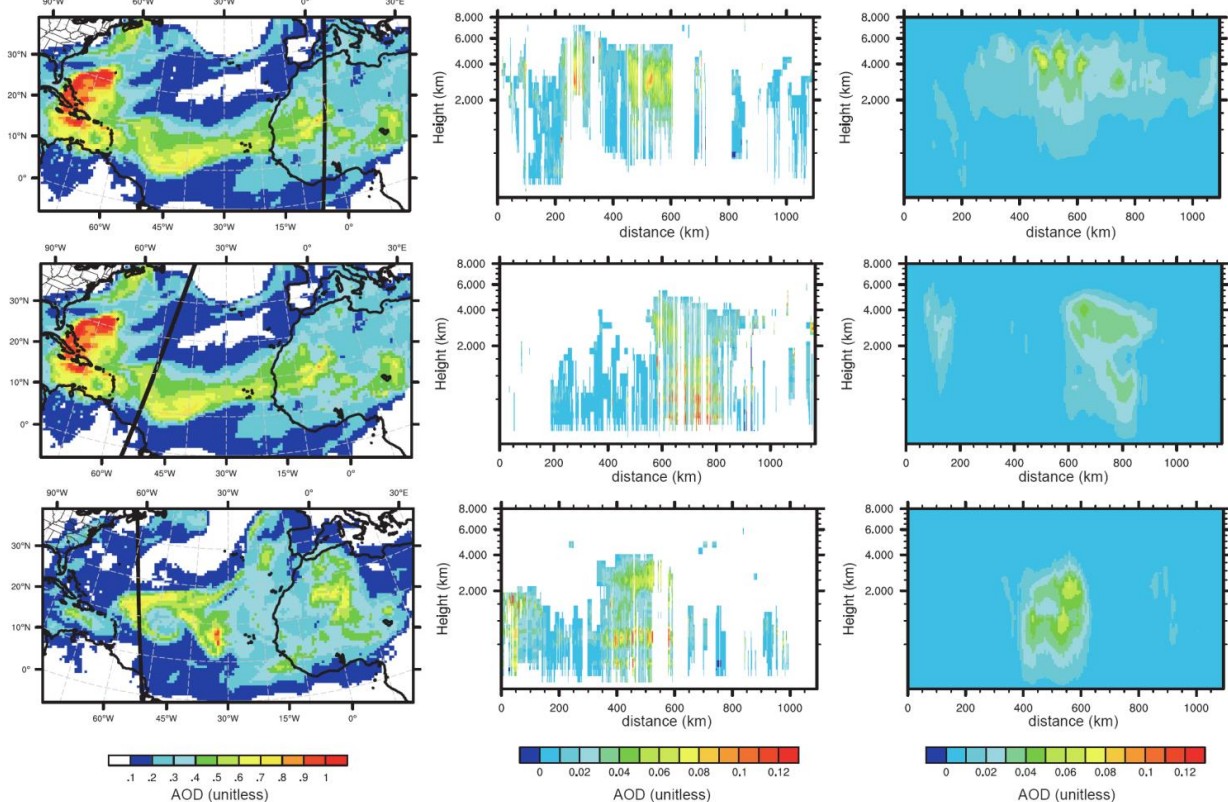

**Figure 4.** Model 550-nm AOD forecasts (left) overlaid with CALIPSO path (black thick lines) and corresponding vertical distributions of CALIOP AOD (middle) and modeled AOD (right) at around 01:00 UTC (top) and 05:00 UTC (middle) of Jul. 29, and 16:00 UTC Sep. 5 (bottom).





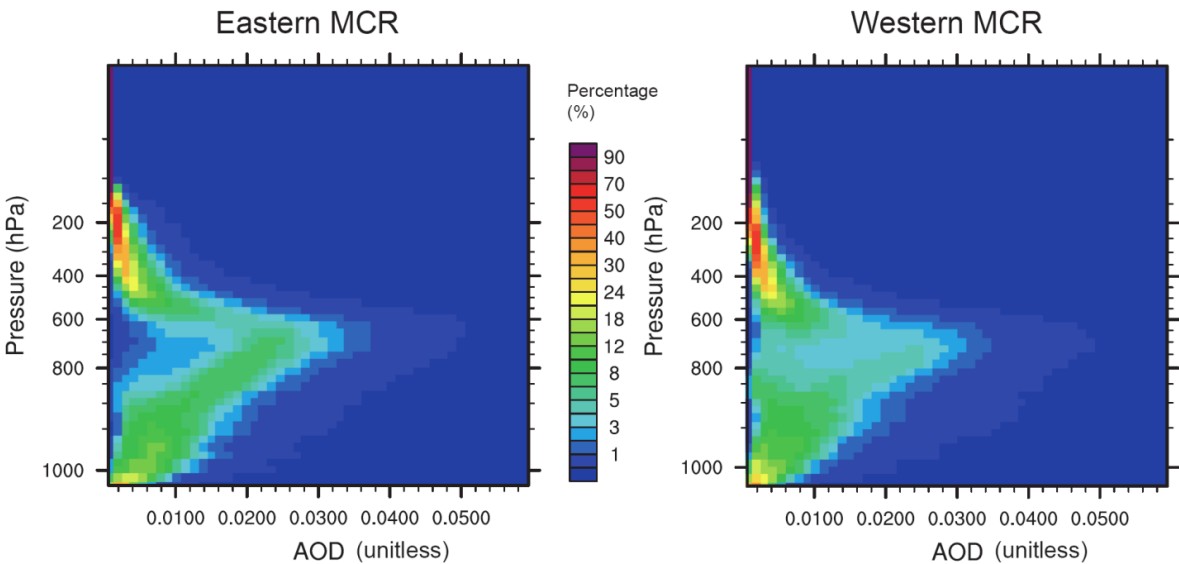

**Figure. 5** AOD CFADs (Contoured Frequency Altitude Diagrams) for the eastern and western MCR regions (based on the first 96 hour forecasts for Jul.8-Sep. 18)





**Figure 6.** Averaged parameters as a function of pressure layers in hPa (y-axis) and the first 96-hour forecast time (x-axis, in UTC time) of the "Elevated Dust" in the eastern MCR from two experiments (Jul. 8 - Sep 18). (a) AOD from RF_ON, and differences (RE_ON - RE_OFF) for temperature (degree), relative humidity (%,), vertical velocity (mm/s), CIN (J/kg) and Buoyancy for boundary layer parcels ($10^{-2}$ m/s$^2$). (b) mixing ratios of clouds (liquid plus ice water) (mg/kg); the RE_ON - RE_OFF differences of ice clouds were given as contour lines (negative values in dash lines). (c) the differences of heating rates (K/day) due to LW, SW and LW+SW radiative effects respectively. Black dots indicate the top of boundary layer.





**Figure 7.** Same as Fig. 6 but for the "Elevated Dust" in the western MCR.





**Figure 8.** Same as Fig. 6 but for the "Deep Layer of Dust" in the eastern MCR.





**Figure 9.** Same as Fig. 6, but for the "Deep Layer of Dust" in the western MCR.



**Figure 10.** Aerosol-induced Sep. 5 suppression case of TCs. The four rows are 10-m u-v wind vectors overlaid with potential temperatures at 900 hPa for RE_ON (left) and RE_OFF (right) at the forecast time. The red rectangles are the GFDL tracked cyclone regions. The right panel is the skew-T/log-P diagrams for the cyclone region average (red: RE_ON, blue: RE_OFF).





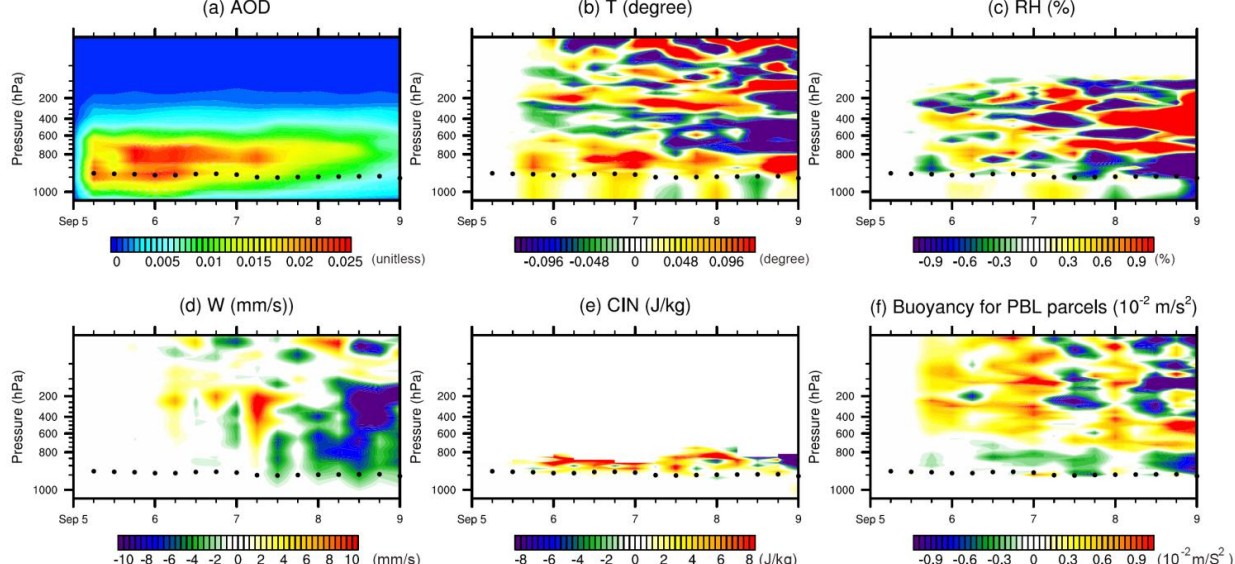

**Figure 11.** Tracked-cyclone-area averaged parameters (as a function of height and time) from two experiments for the forecast initiated from 00UTC 5 Sep: (a) AOD from RE_ON, and differences (RE_ON - RE_OFF) for (b) temperature (degree), (c) relative humidity (%,), (d) vertical velocity (mm/s), (e) CIN (J/kg) and (f) Buoyancy for boundary layer parcels ($10^{-2}$ m/s$^2$). The cyclone area is tracked by GFDL Tracker and shown as the red rectangles in Fig. 10. Black dots indicate the top of boundary layer.



**Figure 12.** West-East cross sections of the cycle area (along red center in Fig. 10) of (u, 100×W) streamline and temperatures from the (a) RE_ON and (b) RE_OFF experiments as well as (c) the corresponding differences (RE_ON – RE_OFF). (d) is relative humidity difference of RE_ON minus RE_OFF. The two experiments are for 78-hr forecast valid at 06UTC 8 Sep (initiated from 00 UTC 5 Sep). The suppressed disturbance center was at 44.6 °W.



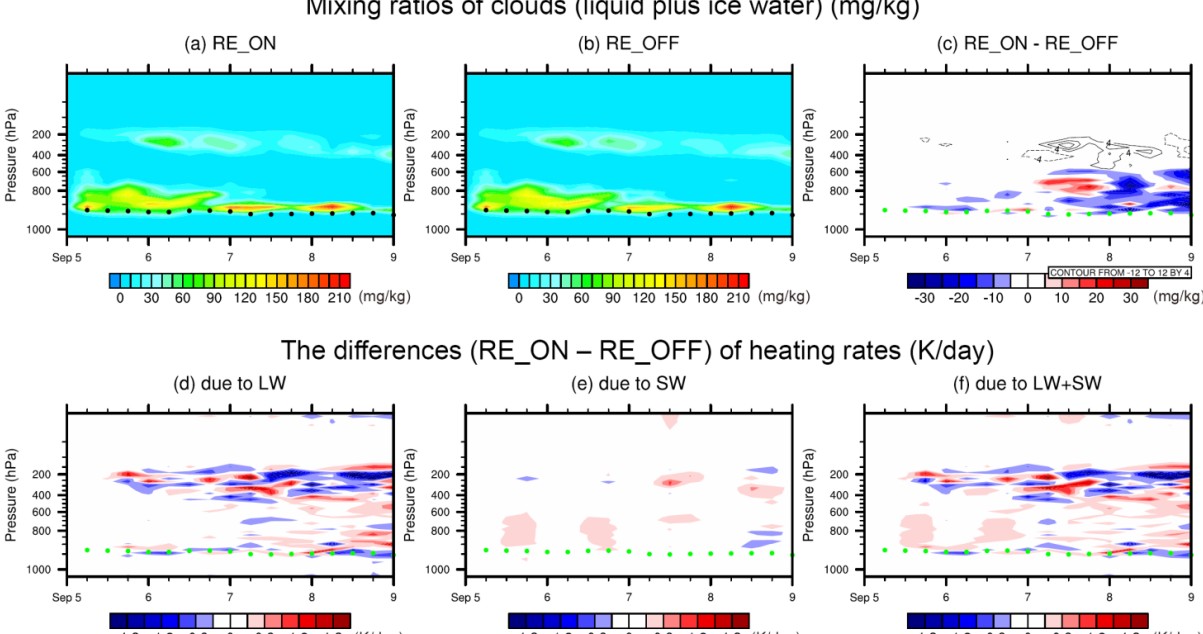

**Figure 13.** Same as Fig. 11, but for different parameters. (a) and (b) are mixing ratios of clouds (liquid plus ice water) (mg/kg) from RE_ON and RE_OFF respectively; (c) is RE_ON - RE_OFF differences of liquid clouds (shaded) and ice clouds (contour lines, negative values in dash lines); (d)-(f) are the differences of heating rate (K/day) due to LW (d), SW (e) and LW+SW (f) radiative effects respectively. Black/green dots indicate the top of boundary layer.