# Peer review of "Dust Radiative Effects on Atmospheric Thermodynamics and Tropical Cyclogenesis over the Atlantic Ocean Using WRF/Chem Coupled with an AOD Data Assimilation System"

_Atmospheric Chemistry and Physics, 2016_

## Referee Comment (RC1) · Anonymous Referee #1 · 3 Dec 2016

This paper discusses the direct radiative effects of dust on atmospheric thermodynamics and tropical cyclogenesis during a Saharan dust outbreak in the summer 2006 in two different regions over the Atlantic Ocean in a modeling study using the WRF/Chem model coupled with a data assimilation system of aerosol optical depth. This study is interesting and novel in design. It addresses an important question of how dust interacts with tropical cyclones because that is still a matter of debate. Dust is one factor that is not taken into account in the seasonal TC forecasts and on hindsight has been made responsible for the rather inactive 2006 Atlantic TC season. The topic of the paper is appropriate for ACP.

The authors first analyzed all 4-day forecasts from Jul 8 to Sept 18, 2006 once with the radiative effects of dust switched on and once switched off. More specifically they looked at the hourly results of simulated dust in the eastern vs. western main cyclogenesis region (MCR) and further divided the simulated results into elevated dust and deep-layer dusts. They find that deep-layer dust that is located at low altitudes inhibits convection because it increases convective inhibition through radiative heating of dust immediately above the boundary layer. They found the opposite for elevated dust. I am not sure that I agree with their findings because the paper is rather cumbersome to read. The results between the radiative effects of dust on (RE_ON) and dust off (RE_OFF) are rather ambiguous, sometimes not supported or poorly explained and some of their results may be overinterpreted. Instead of just showing time series of the differences between RE_ON and RE_OFF, a statistical analysis would be needed in order to convince the readers that their results are statistically significant and not within the noise. If this aspect and the detailed comments listed below are addressed, I recommend the paper for publication in ACP.

Detailed comments:

p.2, l. 24/25: The sentence seems wrong, what is meant by "as in semi-direct effect", as consistent with the semi-direct effect?

p.3, l.29: radiatively active

p.6, l. 6: why didn't you also diagnose AOD at 483.5 nm from the model?

p.7, l. 13: how did you come up with the definition of "deep layer of dust".? Is there a reference or justification for this? If this definition of dust gives you only 8.5-10.7% of the cases, and the rest of the cases are elevated dust, then you have 10 times as many cases of elevated dust. That doesn't seem to be appropriate. Or maybe you need to specify better what you mean by total sampling ratios.

p.7, l. 33: I disagree, the positive RH anomaly goes along with a negative T anomaly

just above the PBL, not a positive one. Higher up both anomalies have the same sign, how do you explain that?

p.7, l.36: I disagree: how can there be reduced radiation below the dust layer if the SW radiation anomalies are positive?

p.8, l. 1: How would ice clouds play a role? Explain.

p.8, l.9: Why is the warming of the dust stronger if elevated? Explain.

p.8, l.30: There is no decrease in the net heating rate below 900 hPa at least no significant one. If you refer to the LW heating that seems to be too small to matter. Please correct.

p.8, l.31: Why do ice clouds change? Explain.

p.8, l.36: What about changes in adiabatic expansion and cooling? If you show heating rates, it would be great if you looked at all the contributions, i.e. add heating rates from phase changes, vertical diffusion and advection in order to understand your results.

p.9, l.8/9: No statistics are presented, just differences of time series. For statistics it would be necessary that you calculated the significance of the changes in RE_ON - RE_OFF as compared to natural variability or something like this.

p.9, l.30: ...air is relatively dry.... For what?

p.9, l.32: decreases

p.11, line 2: Figure 13d is so patchy that I don't agree that there is a significant warming effect at cloud top. That is not convincing.

---

## Referee Comment (RC2) · Anonymous Referee #2 · 11 Mar 2017

Chen et al. presented a study focusing on mineral dust radiative effects on thermo-dynamics and tropical cyclogenesis over the Atlantic Ocean using the meteorology-chemistry coupled WRF-Chem model. A 3DVAR data assimilation system is employed to assimilate MODIS aerosol optical depth (AOD) data to improve model simulation of dust distribution. The study investigated the impacts of Saharan dust layers over the North Atlantic Ocean near the source region and the region further downwind. A main finding is that mineral dust layers either enhance or suppress convection depending on their location relative to the boundary layer.

[Figure]

Quantifying the dust radiative effects of tropical cycolgenesis is a challenging issue and is of great scientific importance. The study can reduce some of the uncertainties with the help of an aerosol data assimilation system. I have some comments listed below mainly for clarification, which I think the authors shall address before considering publish.

**Specific Comments**
1) Page 4, Line 2:
"Statistics based on long-term periods . . .". The word "long-term" can be misleading. Suggest change to "summer 2006" that the study was focused on.

2) Page 6, Line 15-20:
Suggest report the statistics (e.g., mean, correlation coefficients) for the comparisons shown in Figure 3.

3) Page 7, Line 10-13:
The definition of "Deep Layer of Dust" is confusing. Its difference with "Elevated Dust" is not clear in the text. From Figure 8 and 9, it appears that the "deep layer of dust" shows concentrated AOD below 800 hPa while also extending to the free troposphere. Please clarify.

4) Page 7, Line 25-30:
Can you explain why there are significant positive temperature anomalies in the dust layer and negative temperature anomalies above and below the dust? I was expecting some negative temperature anomalies because dust is also radiative scattering.

5) Page 8, Line 4-5:
Please provide a definition of "buoyancy for PBL parcels", and explain why they are negatively correlated with the temperature anomalies.

[Figure]

6) Page 8, Line 9-20:

For the "Elevated Dust" in the western region category, again it is not clear to me why the direct (scattering) and semi-direct (absorption) effects of dust could lead to such a strong warming. Please explain in the text.

7) Page 10, Line 10-15:

What does the right panel of Figure 10 show? The dashed line vs. solid line? This should be explained in the text as well as in the figure caption.

8) Page 10, Line 15-18:

In Figure 11, why are there large temperature perturbations in the middle and upper troposphere? Through any mechanical pathway or numerical noises? Please clarify.

9) Page 10, Line 21-22:

Figure 12c-d show the differences of temperature and RH due to associated with the dust radiative effects. How are the difference patterns linked to the case study? This is not described in the text.

10) Page 26, Figure 10:

Please state in the caption what the red dot represents.

---

## Author Comment (AC1) · 23 May 2017

**Reviewer 1#**
This paper discusses the direct radiative effects of dust on atmospheric thermodynamics and tropical cyclogenesis during a Saharan dust outbreak in the summer 2006 in two different regions over the Atlantic Ocean in a modeling study using the WRF/Chem model coupled with a data assimilation system of aerosol optical depth. This study is interesting and novel in design. It addresses an important question of how dust interacts with tropical cyclones because that is still a matter of debate. Dust is one factor that is not taken into account in the seasonal TC forecasts and on hindsight has been made responsible for the rather inactive 2006 Atlantic TC season. The topic of the paper is appropriate for ACP.

The authors first analyzed all 4-day forecasts from Jul 8 to Sept 18, 2006 once with the radiative effects of dust switched on and once switched off. More specifically they looked at the hourly results of simulated dust in the eastern vs. western main cyclogenesis region (MCR) and further divided the simulated results into elevated dust and deep-layer dusts. They find that deep-layer dust that is located at low altitudes inhibits convection because it increases convective inhibition through radiative heating of dust immediately above the boundary layer. They found the opposite for elevated dust. I am not sure that I agree with their findings because the paper is rather cumbersome to read. The results between the radiative effects of dust on (RE_ON) and dust off (RE_OFF) are rather ambiguous, sometimes not supported or poorly explained and some of their results may be overinterpreted. Instead of just showing time series of the differences between RE_ON and RE_OFF, a statistical analysis would be needed in order to convince the readers that their results are statistically significant and not within the noise. If this aspect and the detailed comments listed below are addressed, I recommend the paper for publication in ACP.

We thank Referee # 1 for their comments and suggestions that have helped to improve this manuscript. Our responses to comments and the corresponding changes to the manuscript are detailed below in blue text. Revised manuscript is after the response letter.

The reviewer's comments have been well taken. Specifically, to show statistical significance of our results, we have added the CFAD (Contoured Frequency Altitude Diagrams) and histogram statistics of the hourly results from the 4-day forecasts in addition to the averaged differences of time series (in Fig. 6-9d). Those can serve as evidence that our results are statistically significant. Besides, we also investigated the thermodynamic budget and checked all the contributors in addition to the radiative heating rates (in Fig. 6-9e). To explain why dust induced heating lead to changes vertically far from the source, we raised our assumption that dust-induced heating played as a forcing that caused effects far from the source through upward propagating gravity waves. Finally, we want to emphasize that the purpose of our manuscript is to show the dust radiative effects on atmospheric thermodynamics. There are still a lot of

issues/questions that we can't answer from those statistics due to the complexity of this topic. To be scientifically solid, we have removed some of our original assumptions.

Detailed comments:

**C1.** p.2, l. 24/25: The sentence seems wrong, what is meant by "as in semi-direct effect", as consistent with the semi-direct effect?

We have corrected the sentence. "All these lead to either a positive or negative radiative forcing, as in semi-direct effect (Gu et al., 2010, 2015), which is poorly understood for Saharan dust." ->  All these lead to either a positive or negative radiative forcing associated with semi-direct effect (Gu et al. 2010, 2015), which is poorly understood for Saharan dust

**C2.** p.3, l.29: radiatively active

Corrected

**C3.** p.6, l. 6: why didn't you also diagnose AOD at 483.5 nm from the model?

We have only output the 550-nm AOD from the model, to compare with the MODIS 550-nm AOD data. WRF-Chem has no diagnostic output for 483.5nm AOD.

**C4.** p.7, l. 13: how did you come up with the definition of "deep layer of dust".? Is there a reference or justification for this? If this definition of dust gives you only 8.5-10.7% of the cases, and the rest of the cases are elevated dust, then you have 10 times as many cases of elevated dust. That doesn't seem to be appropriate. Or maybe you need to specify better what you mean by total sampling ratios.

The purpose to distinguish "the Elevated dust" and the "Deep layer of dust" is to investigate the different radiative effects of dusts at different altitudes. We agree that the names of "Deep Layer of Dust" is not very appropriate and we have clarified in the text that "Deep Layer" actually means that the high AOD are mostly located below 700 mb (not really deep). As summarized in the introduction, the semi-direct effect of dust is sensitive to the position of the dust layer relative to clouds. While the definitions of "Elevated Dust" and "Deep Layer of Dust" are somewhat arbitrary, we tried to distinguish the different vertical structures of dusts in the eastern and western MCR regions. As we can see from the CFADs of AOD in Fig. 5, in the eastern MCR the vertical distribution of dusts in most grids is in a rotated V shape, where the high AOD are almost elevated around 700 hPa ; in the western MCR, high AOD can be frequently

found at 900-950 hPa. To be consistent in the two regions and to distinguish the two types of structures, we define "Deep Layer of Dust" as the case when AOD is larger than 0.015 at 900 or 950 hPa but smaller than 0.02 at 700 hPa. When sampling, we checked all the grids/columns within the region for the 96-hr forecasts of Jul. 8- Sep. 18, 2006, when it is qualified for the "Deep Layer of Dust" criteria, it is recorded. The total sampling ratios means the numbers of grids that qualify for the criteria to the numbers of total grids. The remaining cases are defined as the "Elevated Dust" while they show the common characteristic of elevated dust (high AOD elevated at 700 hPa).

**C5.** p.7, l. 33: I disagree, the positive RH anomaly goes along with a negative T anomaly just above the PBL, not a positive one. Higher up both anomalies have the same sign, how do you explain that?

The reasons for the negative temperature anomalies below and above the dust are mainly from the aspect of atmospheric dynamics, and also the changes of RH.  In brief, it's similar to the internal gravity wave that the dust-induced heating played as a forcing that caused effects far from the source through upward propagating gravity waves. This is a possible paradigm shift in the way we think of the total response from a direct effect of aerosols when the aerosol has a mesoscale organizational structure.

To understand this question, we want to raise the concept of gravity wave response induced by absorbing-dust caused periodic heating.  As another reviewer is an expert on atmospheric research with limited background in dynamics, we have put some basis of gravity wave here. Please jump to the next session (separate by  ==========) if you are familiar with this part.

Figure S1(a)-(c) shows the basis of gravity waves. Figure S1(a) is the typical example of gravity wave that illustrated by the wave that occurred on lakes after small disturbance. Figure S1(b) illustrates the buoyancy oscillation in the atmosphere, that when parcel moves vertically from the equilibrium height $(z = z_0)$ after a disturbance, adiabatic buoyancy oscillation would occur due to the buoyancy forcing which would constrain the parcel back to the equilibrium height. The propagation of the buoyancy oscillation leads to internal gravity wave as shown in Fig. S1(c). Figure S1(a)-(c) and the equations are referred to the book "Dynamic Meteorology" (Lv, M., Hou. Z., Zhou Y., Dyanmic Meteorolgy, China Meteorological Press, 2004. P 195-210).

[Figure]

Figure S1. (a) Gravity wave propagation on a lake, (b) Buoyancy oscillation, (c) The propagation of buoyancy oscillation, (d) The propagation of gravity wave induced by absorbing-dust caused heating.

In the case of buoyance oscillation, following equations are given

$$\frac{\mathrm{d}\omega}{\mathrm{d}t} = \frac{\mathrm{d}^2}{\mathrm{d}t^2}(\delta z) = -\frac{1}{\rho}\frac{\partial p}{\partial z} - g,$$

$\omega$ is the parcel vertical velocity, $p$ and $\rho$ are the pressure and density of the parcel, $g$ is the gravity velocity. If the hydrostatic equilibrium is fulfilled by the environment pressure, then

$$\frac{\partial \overline{p}}{\partial z} = -\overline{\rho}g,$$

$\overline{p}$ and $\overline{\rho}$ are the pressure and density of the environment. The parcel pressure is equilibrium to the environment pressure, $p = \overline{p}$ , thus the vertical motion can be given as

$$\frac{d^2}{dt^2}(\delta z) = g\left(\frac{\overline{\rho}-\rho}{\rho}\right),$$

According to the definition of potential temperature, then the equation can be given as

$$\frac{d^2}{dt^2}(\delta z) = g\left(\frac{\theta-\overline{\theta}}{\overline{\theta}}\right),$$

$\overline{\theta}$ is the potential temperature of environment. Assume that $\theta_{(z_0)} = \overline{\theta}_{(z_0)}$ at the equilibrium height $(z = z_0)$, then the potential temperature of environment at height $z_0 + \delta z$ can be written as

$$\overline{\theta}_{(z_0+\delta z)} \cong \overline{\theta}_{(z_0)} + \left(\frac{d\overline{\theta}}{dz}\right)\delta z,$$

As the motion is adiabatic and the potential temperature is conserved for the parcel, thus

$$\theta_{(z_0+\delta z)} = \overline{\theta}_{(z_0)},$$

And the equation can be written as

$$\frac{d^2}{dt^2}(\delta z) + N^2\delta z = 0,$$

And

$$N^2 \equiv g\frac{d\ln\overline{\theta}}{dz},$$

The adiabatic buoyancy oscillation would occur around the equilibrium height for the parcel, and the frequency of the oscillation is $N$. $N$ is called as the Brunt-$V\ddot{a}is\ddot{a}l\ddot{a}$ frequency.

Figure S1 (c) shows how the buoyancy oscillation is propagated. In the figure, $\delta$ is the horizontal divergence. As the buoyancy oscillation occurs in column AA' and the vertical velocity at the boundary $\omega = 0$, there would be horizontal convergence in the bottom of AA' and divergence in the top. The convergence and divergence in column AA' would cause the opposite motions in originally undisturbed AB and A'B' columns, which lead to subsidence at $z = z_0$ in AB and A'B' columns. Thus the vertical motion in column AA' is propagated to adjacent columns by horizontal convergence and divergence. In addition the disturbances of horizontal velocities in the bottom and top regions changes

alternatively, thus it shows wave status vertically. As we can see the internal gravity wave propagates both horizontally and vertically.

===============================

Similar to the buoyancy oscillation illustrated in Figure S1(c), our assumption of the gravity wave response induced by absorbing-dust caused periodic heating is shown in Figure S1(d). The typical three dimensional perturbation equations in the atmosphere are as below.

$$
\begin{cases}
\dfrac{\partial u'}{\partial t} - f_0 v' + \dfrac{\partial}{\partial x}\left(\dfrac{p'}{\rho_0}\right) = 0 \\[2mm]
\dfrac{\partial v'}{\partial t} + f_0 u' + \dfrac{\partial}{\partial}\left(\dfrac{p'}{\rho_0}\right) = 0 \\[2mm]
\lambda \dfrac{\partial \omega'}{\partial t} + \dfrac{\partial}{\partial z}\left(\dfrac{p'}{\rho_0}\right) - \dfrac{\theta'}{\overline{\overline{\theta}}} g = 0 \\[2mm]
\dfrac{\partial u'}{\partial x} + \dfrac{\partial v'}{\partial y} + \dfrac{\partial \omega'}{\partial z} = 0 \\[2mm]
\dfrac{\partial}{\partial t}\left(\dfrac{\theta'}{\overline{\overline{\theta}}}\right) + \dfrac{N^2}{g} \omega' = 0
\end{cases}
$$

When considering the absorbing-dust caused periodic heating force, then the right term in the last energy equation is not zero anymore and it can be written as

$$
\frac{\partial}{\partial t}\left(\frac{\theta'}{\overline{\overline{\theta}}}\right) + \frac{N^2}{g}\omega' = Q_{(x,y,z)} \times \sin(w*t),
$$

$Q_{(x,y,z)}$ is the heating function that associated with the dust position $(x,y,z)$ and also time. It has diurnal cycle which is presented as $\sin(w*t)$ .

Compared to the typical internal gravity wave that caused by buoyancy oscillation, the only difference is the dust-induced heating force is much stronger and last for a few hours, thus the wave propagation is also stronger. Besides, it's a periodic forcing with diurnal cycle that the vertical patterns changes alternatively during the daytime (with heating force) and nighttime (without heating source). We believe that most everything we see can be explained by a theoretical treatment of the linear primitive equations forced by a period heat source. Within the heating, upward motion will increase water vapor, but adiabatic cooling will not quite offset the heating, so you can have T and q in phase. But this will not be the case above and below the heating because there is no heating to balance adiabatic changes. Furthermore, because waves are excited in the stratified background state, they propagate vertically and influence temperature and RH changes far from the source (as illustrated in Fig. S1(d)). In particular, I think that is what is happening to the ice clouds. The upward motion from the waves increases the nucleation

of ice; subsidence decreases it. As the period heat source occurs every 24 hour and last for a few hours, the propagation of the gravity wave strengthens horizontally and vertically as time increase. I believe that a simple treatment of linear waves could explain a great deal of what we see. While it's our assumption and more evidence should be addressed in our future study. We have summarized our assumptions in the manuscript (Page 8, lines 25-38)

**C6.** p.7, l.36: I disagree: how can there be reduced radiation below the dust layer if the SW radiation anomalies are positive?

Thanks! Corrected in the text "As the clouds fields (middle panels) show no significant low-cloud changes and heating rates changes due to LW (bottom panels) at 800-950 hPa layer, the cooling below 800 hPa may result from the adiabatic upward motion driven by the heating above".

**C7.** p.8, l. 1: How would ice clouds play a role? Explain.

It appears that ice cloud changes are responses to the vertical motion induced by the heating below. Please see the details in the answers to C5.

**C8.** p.8, l.9: Why is the warming of the dust stronger if elevated? Explain.

We want to say "For the "Elevated Dust" in the western MCR (Fig. 7), the warming due to dust is stronger than that in the eastern MCR". We didn't emphasize that the warming of the dust is stronger if elevated.

We have tried to explain the reason in the text why the heating due to dust in the western MCR is much stronger than that in the eastern MCR for the "Elevated Dust". "The differences of the temperature anomalies above the boundary layer in the two regions may come from the different clouds and heating rates changes (Fig. 6b,c and 7b,c). In both the western and eastern MCR, the low clouds are right at the top of boundary layer. In the western MCR, the cloud amounts are larger than that in the eastern MCR (as the air becomes moister from eastern to western MCR) and the cloud changes are also more prominent. As the dust is concentrated immediately above the low clouds, it is possible that the cloud concentrations are decreased at 900-950 hPa due to the evaporation effect. As the longwave radiation of dust is not included in the model, the heating rates changes due to LW are mainly from the changes of clouds. As cloud-top cooling and cloud-bottom heating normally occur associated with longwave (LW) radiative effects (Fu and Liou 1993), the vertical changes of clouds cause significant positive heating rates

anomalies at 900-950 hPa for the western MCR (Fig. 7c) that offset the adiabatic cooling there."

We also showed the temperature tendency differences from all the contributions, including radiative heating, vertical mixing from PBL process, cumulus process, diabatic heating, total advective processes and the sum of the five parameters. The Figures S2 and S3 (in the answers to C11 as below) indicate our assumption. The cloud changes in the western MCR are important and thus changes the temperature tendencies anomalies among different processes. In the western MCR, the negative temperature tendency differences due to advective process (Fig. S3) are much weaker than that in the eastern MCR (Fig. S2) and thus the positive temperature anomalies are much larger than that in the eastern MCR.

**C9.** p.8, l.30: There is no decrease in the net heating rate below 900 hPa at least no significant one. If you refer to the LW heating that seems to be too small to matter. Please correct.

Thanks! We are referring to the LW heating decrease and we have corrected it in the text. "The changes in low level clouds lead to an increase in heating rate around 900 hPa and a slight LW decrease below 900 hPa."

**C10.** p.8, l.31: Why do ice clouds change? Explain.

Please see the details in the answers to C5. In brief, the changes vertically far from the dust are the responses of the propagation of the gravity wave and the root drive is the dust-induced heating force. The upward motion from the waves increases the nucleation of ice; while subsidence decreases it. While it's still our assumption and more evidence should be addressed in the future study.

**C11.** p.8, l.36: What about changes in adiabatic expansion and cooling? If you show heating rates, it would be great if you looked at all the contributions, i.e. add heating rates from phase changes, vertical diffusion and advection in order to understand your results.

Thanks for the suggestion! Figure S2-S5 (same as Fig. 6-9e in the manuscript) show the temperature tendency differences from all the contributions, including radiative heating, vertical mixing from PBL process, cumulus process, diabatic heating, total advective processes and the sum of the five parameters. The four figures are for two types of dusts ("Elevated Dust" and "Deep Layer of Dust") in the two regions (the eastern and western MCR) respectively. The only contribution that's not readily available from the model is the heating term due to horizontal diffusion which would probably be part of a residual.

The term from total advective processes includes both horizontal advection and vertical advection. As horizontal advection becomes less important as the box size increases, the term of total advective process mainly reflects the vertical advection, including the vertical transport of heat (temperature) and the adiabatic cooling. Condensational or phase change term, can be expressed as the sum of cumulus and diabatic heating.

From Fig. S2-S3, we can see that for "Elevated Dust", the major contributions of temperature tendency differences are radiative and advective processes. As we discussed in the answers to C5, gravity wave response caused by absorbing-dust induced heating is expected and here the tendency changes due to advective process indicated our assumption.

[Figure]

Figure S2. Period averaged (Jul. 8 - Sep 18) temperature tendency differences (RE_ON - RE_OFF) from all contributions, as a function of pressure layers in hPa (y-axis) and the first 96-hour forecast time (x-axis, in UTC time) of the "Elevated Dust" in the eastern MCR. (a) Radiative process, (b) Vertical mixing by PBL process, (c) Cumulus process, (d) Diabatic heating, (e) Total advective process, and (f) the sum of the five processes. Units: (K/day). Black dots indicate the top of boundary layer.

[Figure]

Figure S3. Same as Fig. S2 but for the "Elevated Dust" in the western MCR.

Compared to the "Elevated Dust", in the case of "Deep Layer of Dust" in the eastern MCR (Fig. S4), the contributions from vertical mixing from PBL process, cumulus and diabatic heating are also important, revealing that phase changes take place. We have added a few discussions in the manuscript too.

[Figure]

Figure S4. Same as Fig. S2 but for the "Deep Layer of Dust" in the eastern MCR.

[Figure]

Figure S5. Same as Fig. S2 but for the "Deep Layer of Dust" in the western MCR.

We have put those figures in Fig.6-9e and also added a few discussions in the manuscript (Page 7, lines 32-39; Page 8, lines 18-24; Page 9, lines 19-21; Page 10, lines 1-6).

**C12.** p.9, l.8/9: No statistics are presented, just differences of time series. For statistics it would be necessary that you calculated the significance of the changes in RE_ON - RE_OFF as compared to natural variability or something like this.

Thanks for this suggestion! Yes, while the averaged differences are significant, there are also larger changes that occur within the averaging area. To illustrate both aspects, we have added the CFADs (Contoured Frequency Altitude Diagrams) of AOD and the differences (RE_ON – RE_OFF) of temperature, for the "Elevated Dust" and "Deep Layer of Dust" in the eastern and western MCR respectively (Fig. S6-S9, same as Fig. 6-9d in the manuscript). The histogram and statistics (mean value and standard deviation) of temperature differences at certain levels are also given.

In Figure S6 (Statistics of "Elevated Dust" in the eastern MCR), we can see the frequency figures at 36-hr and 48-hr forecasts are consistent with the time series, in most grids it instantly show positive T anomaly at dust layer (around 700 hPa) during the daytime and it shows negative T anomaly below dust layer (around 900 hPa) at 48-hr forecast. The histograms of temperature differences at 730 hPa (for 36-hr forecast) and at 915 hPa (for 48-hr forecast) further reveal that those results are not within noise. Specifically, the

statistics of temperature difference at 915 hPa (at 48-hr forecast) show the mean value is negative (-0.044 degree) with standard deviation 0.441 degree.

Similarly in Figure S7 (Statistics of "Elevated Dust" in the western MCR), the maximum differences in some grids are much larger than the averaged values in Fig. 7a. Besides, in the manuscript, we raised one assumption that "the vertical changes of clouds cause significant positive heating rates anomalies at 900-950 hPa for the western MCR (Fig. 7c) that offset the adiabatic cooling there" as no significant T anomaly at 900-950 hPa were shown from the averaged differences in Fig. 7a. While it's interesting that two kinds of T anomaly (negative and positive) did show at 900-950hPa in the CFADs at 48-hr forecast. The CFADs further strengthen our statement.

[Figure]

Figure S6. Statistics for the "Elevated Dust" in the eastern MCR at 36-hr (top) and 48-hr (bottom) forecasts. The first and second columns are the CFADs of AOD and differences (RE_ON-RE_OFF) of temperature respectively. The third columns are the histograms of temperature differences at certain levels (indicated as horizontal red lines in the first and second columns). The vertical red lines in the third columns are the lines of value zero.

For both CFADs and histograms, 100 size bins are assigned for X-axis from the minimum to maximum values.

[Figure]

Figure S7. Same as Figure S6 but for the "Elevated Dust" in the western MCR.

As for the "Deep layer of dusts" in the eastern and western MCR (Fig. S8 and S9), the maximum variations are also much larger than the averages.

[Figure]

Figure S8. Same as Figure S6 but for the "Deep Layer of Dust" in the eastern MCR. Only 36-hr forecast is given.

[Figure]

Figure S9. Same as Figure S8 but for the "Deep layer of Dust" in the western MCR.

We have put those figures at 36-hr forecasts in Fig. 6-9d in the manuscript and we also added a few discussions in the manuscript (Page 7, lines 29-31).

**C13.** p.9, l.30: ...air is relatively dry.... For what?

We are discussing about the possibility of the elevated dust-induced TC intensification in the eastern MCR. As we can see from the statistics of Fig. 6a, CIN just above the PBL

decreases due to the adiabatic cooling below the dust layer which may provide opportunities for TC intensification. However, those statistics are based on averages; for a single TC case, it requires very strict conditions as the two key factors are contradictory. In the eastern MCR, the air is relatively dry as it is close to the dust source region, so the cloud amount is relatively smaller compared to the case in the western MCR (Fig. 6b v.s. Fig. 7b). In one hand, the dryness means no significant clouds-induced heating that could offset the adiabatic cooling, while in the other hand, the dryness also means small chance of TC development.

**C14.** p.9, l.32: decreases

Corrected.

**C15.** p.11, line 2: Figure 13d is so patchy that I don't agree that there is a significant warming
effect at cloud top. That is not convincing

We have removed the statement.

[revised manuscript text omitted]

---

## Author Comment (AC2) · 23 May 2017

**Reviewer 2#**

Chen et al. presented a study focusing on mineral dust radiative effects on thermo-dynamics and tropical cyclogenesis over the Atlantic Ocean using the meteorology-chemistry coupled WRF-Chem model. A 3DVAR data assimilation system is employed to assimilate MODIS aerosol optical depth (AOD) data to improve model simulation of dust distribution. The study investigated the impacts of Saharan dust layers over the North Atlantic Ocean near the source region and the region further downwind. A main finding is that mineral dust layers either enhance or suppress convection depending on their location relative to the boundary layer.

Quantifying the dust radiative effects of tropical cycolgenesis is a challenging issue and is of great scientific importance. The study can reduce some of the uncertainties with the help of an aerosol data assimilation system. I have some comments listed below mainly for clarification, which I think the authors shall address before considering publish.

We thank Referee #2 for their comments and suggestions that have helped to improve this manuscript. Our responses to comments and the corresponding changes to the manuscript are detailed below in blue text. Revised manuscript is after the response letter. We have added following three parts of content in the manuscript:

1. To show statistical significance of our results, we have added the CFAD (Contoured Frequency Altitude Diagrams) and histogram statistics of the hourly results from the 4-day forecasts in addition to the averaged differences of time series (in Fig. 6-9d). Those can serve as evidence that our results are statistically significant. We also added a few discussions in the manuscript (Page 7, lines 29-31).
2. We also investigated the thermodynamic budget and checked other contributors in addition to the radiative heating rates (in Fig. 6-9e), including vertical mixing from PBL process, cumulus process, diabatic heating, and total advective processes. We also added a few discussions in the manuscript (Page 7, lines 32-39; Page 8, lines 18-24; Page 9, lines 19-21; Page 10, lines 1-6).
3. To explain why dust induced heating lead to changes vertically far from the source, we raised our assumption that dust-induced heating played as a forcing that caused effects far from the source through upward propagating gravity waves. We have stated our assumptions in the answers to Question 4 and also summarized it in the manuscript (Page 8, lines 25-38).

Specific Comments

1) Page 4, Line 2: "Statistics based on long-term periods ... ". The word "long-term" can be misleading. Suggest change to "summer 2006" that the study was focused on.

Corrected.

2) Page 6, Line 15-20: Suggest report the statistics (e.g., mean, correlation coefficients) for the comparisons shown in Figure 3.

Thanks for the suggestion! The statistics are added in Figure 3.

[Figure]

3) Page 7, Line 10-13: The definition of "Deep Layer of Dust" is confusing. Its difference with "Elevated Dust" is not clear in the text. From Figure 8 and 9, it appears that the

"deep layer of dust" shows concentrated AOD below 800 hPa while also extending to the free troposphere. Please clarify.

Yes, we agree that the name of "Deep Layer of Dust" is not very appropriate. We have clarified in the text that "Deep Layer" actually means that the high AOD are mostly located below 700 mb (not really deep).

The purpose to distinguish "the Elevated dust" and the "Deep layer of dust" is to investigate the different radiative effects of dusts at different altitudes. As summarized in the introduction, the semi-direct effect of dust is sensitive to the position of the dust layer relative to clouds. While the definitions of "Elevated dust" and "Deep layer of dust" are somewhat arbitrary, we tried to distinguish the different vertical structures of dusts in the eastern and western MCR regions. As we can see from the CFADs of AOD in Fig. 5, in the eastern MCR the vertical distribution of dusts in most grids is in a rotated V shape, where the high AOD are almost elevated around 700 hPa; in the western MCR, high AOD can be frequently found at 900-950 hPa. To be consistent in the two regions and to distinguish the two types of structures, we define "Deep Layer of Dust" as the case when AOD is larger than 0.015 at 900 or 950 hPa but smaller than 0.02 at 700 hPa. The remaining cases are defined as the "Elevated Dust" while they show the common characteristic of elevated dust (high AOD elevated at 700 hPa).

4) Page 7, Line 25-30: Can you explain why there are significant positive temperature anomalies in the dust layer and negative temperature anomalies above and below the dust? I was expecting some negative temperature anomalies because dust is also radiative scattering.

In the model, for the Saharan dust with such huge load, it is supposed to affect the environment mainly through the direct radiative absorbing effect, thus there are significant positive temperature anomalies in the dust layer. Besides, the semi-direct effect in the manuscript is through the thermodynamics of clouds (as clouds calculation is associated with T/RH in the model).

The reasons for the negative temperature anomalies below and above the dust are mainly from the aspect of atmospheric dynamics. In brief, it's similar to the internal gravity wave that the dust-induced heating played as a forcing that caused effects far from the source through upward propagating gravity waves. This is a possible paradigm shift in the way we think of the total response from a direct effect of aerosols when the aerosol has a mesoscale organizational structure.

To understand this question, we want to raise the concept of gravity wave response induced by absorbing-dust caused periodic heating. Figure S1(a)-(c) shows the basis of

gravity waves. Figure S1(a) is the typical example of gravity wave that illustrated by the wave that occurred on lakes after small disturbance. Figure S1(b) illustrates the buoyancy oscillation in the atmosphere, that when parcel moves vertically from the equilibrium height $(z = z_0)$ after a disturbance, adiabatic buoyancy oscillation would occur due to the buoyancy forcing which would constrain the parcel back to the equilibrium height. The propagation of the buoyancy oscillation leads to internal gravity wave as shown in Fig. S1(c). Figure S1(a)-(c) and the equations are referred to the book "Dynamic Meteorology" (Lv, M., Hou. Z., Zhou Y., Dyanmic Meteorolgy, China Meteorological Press, 2004. P 195-210).

[Figure]

(a)

(b)

(c)

(d)

Figure S1. (a) Gravity wave propagation on a lake, (b) Buoyancy oscillation, (c) The propagation of buoyancy oscillation, (d) The propagation of gravity wave induced by absorbing-dust caused heating.

In the case of buoyance oscillation, following equations are given

$$\frac{d\omega}{dt} = \frac{d^2}{dt^2}(\delta z) = -\frac{1}{\rho}\frac{\partial p}{\partial z} - g,$$

$\omega$ is the parcel vertical velocity, $p$ and $\rho$ are the pressure and density of the parcel, $g$ is the gravity velocity. If the hydrostatic equilibrium is fulfilled by the environment pressure, then

$$\frac{\partial \bar{p}}{\partial z} = -\bar{\rho}g,$$

$\bar{p}$ and $\bar{\rho}$ are the pressure and density of the environment. The parcel pressure is equilibrium to the environment pressure, $p = \bar{p}$ , thus the vertical motion can be given as

$$\frac{d^2}{dt^2}(\delta z) = g(\frac{\bar{\rho}-\rho}{\rho}),$$

According to the definition of potential temperature, then the equation can be given as

$$\frac{d^2}{dt^2}(\delta z) = g(\frac{\theta-\bar{\theta}}{\bar{\theta}}),$$

$\bar{\theta}$ is the potential temperature of environment. Assume that $\theta_{(z_0)} = \bar{\theta}_{(z_0)}$ at the equilibrium height $(z = z_0)$, then the potential temperature of environment at height $z_0 + \delta z$ can be written as

$$\bar{\theta}_{(z_0+\delta z)} \cong \bar{\theta}_{(z_0)} + (\frac{d\bar{\theta}}{dz})\delta z,$$

As the motion is adiabatic and the potential temperature is conserved for the parcel, thus

$$\theta_{(z_0+\delta z)} = \bar{\theta}_{(z_0)},$$

And the equation can be written as

$$\frac{d^2}{dt^2}(\delta z) + N^2\delta z = 0,$$

And

$$N^2 \equiv g\frac{\mathrm{d}\ln\overline{\theta}}{\mathrm{d}z},$$

The adiabatic buoyancy oscillation would occur around the equilibrium height for the parcel, and the frequency of the oscillation is $N$. $N$ is called as the Brunt-$V\ddot{a}is\ddot{a}l\ddot{a}$ frequency.

Figure S1 (c) shows how the buoyancy oscillation is propagated. In the figure, $\delta$ is the horizontal divergence. As the buoyancy oscillation occurs in column AA' and the vertical velocity at the boundary $\omega = 0$, there would be horizontal convergence in the bottom of AA' and divergence in the top. The convergence and divergence in column AA' would cause the opposite motions in originally undisturbed AB and A'B' columns, which lead to subsidence at $z = z_0$ in AB and A'B' columns. Thus the vertical motion in column AA' is propagated to adjacent columns by horizontal convergence and divergence. In addition the disturbances of horizontal velocities in the bottom and top regions changes alternatively, thus it shows wave status vertically. As we can see the internal gravity wave propagates both horizontally and vertically.

Similar to the buoyancy oscillation illustrated in Figure S1(c), our assumption of the gravity wave response induced by absorbing-dust caused periodic heating is shown in Figure S1(d). The typical three dimensional perturbation equations in the atmosphere are as below.

$$\begin{cases} \dfrac{\partial u'}{\partial t} - f_0 v' + \dfrac{\partial}{\partial x}\left(\dfrac{p'}{\rho_0}\right) = 0 \\[2mm] \dfrac{\partial v'}{\partial t} + f_0 u' + \dfrac{\partial}{\partial}\left(\dfrac{p'}{\rho_0}\right) = 0 \\[2mm] \lambda\dfrac{\partial \omega'}{\partial t} + \dfrac{\partial}{\partial z}\left(\dfrac{p'}{\rho_0}\right) - \dfrac{\theta'}{\overline{\overline{\theta}}}g = 0 \\[2mm] \dfrac{\partial u'}{\partial x} + \dfrac{\partial v'}{\partial y} + \dfrac{\partial \omega'}{\partial z} = 0 \\[2mm] \dfrac{\partial}{\partial t}\left(\dfrac{\theta'}{\overline{\overline{\theta}}}\right) + \dfrac{N^2}{g}\omega' = 0 \end{cases}$$

When considering the absorbing-dust caused periodic heating force, then the right term in the last energy equation is not zero anymore and it can be written as

$$\frac{\partial}{\partial t}\left(\frac{\theta'}{\overline{\overline{\theta}}}\right) + \frac{N^2}{g}\omega' = Q_{(x,y,z)} \times \sin(w * t),$$

$Q_{(x,y,z)}$ is the heating function that associated with the position $(x,y,z)$ and also time. It has diurnal cycle which is presented as $\sin(w * t)$.

Compared to the typical internal gravity wave that caused by buoyancy oscillation, the only difference is the dust-induced heating force is much stronger and last for a few hours, thus the wave propagation is also stronger. Besides, it's a periodic forcing with diurnal cycle that the vertical patterns changes alternatively during the daytime (with heating force) and nighttime (without heating source). We believe that most everything we see can be explained by a theoretical treatment of the linear primitive equations forced by a period heat source. Within the heating, upward motion will increase water vapor, but adiabatic cooling will not quite offset the heating, so you can have significant positive T anomaly in the dust layer. But this will not be the case above and below the heating because there is no heating to balance adiabatic changes. Furthermore, because waves are excited in the stratified background state, they propagate vertically and influence temperature and RH changes far from the source (as illustrated in Fig. S1(d)). In particular, I think that is what is happening to the ice clouds. The upward motion from the waves increases the nucleation of ice; subsidence decreases it. As the period heat source occurs every 24 hour and last for a few hours, the propagation of the gravity wave strengthens horizontally and vertically as time increase. I believe that a simple treatment of linear waves could explain a great deal of what we see.

Below figure show the temperature tendency differences from all the contributions, including radiative heating, vertical mixing from PBL process, cumulus process, diabatic heating, total advective processes and the sum of the five parameters for the "Elevated Dust" in the eastern MCR. The term from total advective processes includes both horizontal advection and vertical advection. As horizontal advection becomes less important as the box size increases, the term of total advective process mainly reflects the vertical advection, including the vertical transport of heat (temperature) and the adiabatic cooling. We can see that the major contributions of temperature tendency differences are radiative and advective processes. As we discussed above, gravity wave response caused by absorbing-dust induced heating is expected and here the tendency changes due to advective process indicated our assumption. The cooling by advection explains the temperature cooling around 400 mb and the layer below dust.

[Figure]

Figure S2. Period averaged (Jul. 8 - Sep 18) temperature tendency differences (RE_ON - RE_OFF) from all contributions, as a function of pressure layers in hPa (y-axis) and the first 96-hour forecast time (x-axis, in UTC time) of the "Elevated Dust" in the eastern MCR. (a) Radiative process, (b) Vertical mixing by PBL process, (c) Cumulus process, (d) Diabatic heating, (e) Total advetive process, and (f) the sum of the five processes. Units: (K/day). Black dots indicate the top of boundary layer.

5) Page 8, Line 4-5: Please provide a definition of "buoyancy for PBL parcels", and explain why they are negatively correlated with the temperature anomalies.

Buoyancy represents the difference between the pressure-gradient force and gravity for parcels. The difference in temperature between an air parcel and its immediate environment governs the buoyancy of the parcel.

Here the calculation of "buoyancy for PBL parcels" is borrowed from the CAPE (convective available potential energy) subroutine. Generic CAPE is calculated by integrating vertically the local buoyancy of a parcel from the level of free convection (LFC) to the equilibrium level (EL):

$$CAPE = \int_{z_f}^{z_n} g(\frac{T_{v,parcel} - T_{v,env}}{T_{v,env}}) \, dz$$

Where $z_f$ is the height of the level of free convection and $z_n$ is the height of the equilibrium level (neutral buoyancy), where $T_{v,parcel}$ is the virtual temperature of the

specific parcel, where $T_{v,env}$ is the virtual temperature of the environment, and $g$ is the acceleration due to gravity.

CAPE is the integrated energy of a parcel while we want to check the most sensitive layer that at the PBL where very small buoyancy changes might change the direction (negative or positive). Thus we defined the "buoyancy for the parcels at PBL", it is calculated before the integration of distance. For the layers below dust, the temperature anomalies of environment are negatively correlated with buoyancy for PBL parcels.

6) Page 8, Line 9-20: For the "Elevated Dust" in the western region category, again it is not clear to me why the direct (scattering) and semi-direct (absorption) effects of dust could lead to such a strong warming. Please explain in the text.

As we stated in the answers to question (4), the direct effect of Saharan dust is mainly through the radiative heating due to its absorbing characteristic defined in the model. And our focus is on thermodynamic changes that mainly influence area-mean profiles of temperature and moisture due to direct effect. The semi-direct effect is through the thermodynamics of clouds. Although the microphysical influence of dust aerosols acting as cloud condensation nuclei or ice nuclei is not included in the model set up; as the cloud cover calculations in the model are relevant to the RH/T changes, the cloud-induced semi-direct effect is indeed represented as a consequence of the direct effect in the model.

We have tried to explain the reason in the text why the heating due to dust in the western MCR is much stronger than that in the eastern MCR for the "Elevated Dust". "The differences of the temperature anomalies above the boundary layer in the two regions may come from the different clouds and heating rates changes (Fig. 6b,c and 7b,c). In both the western and eastern MCR, the low clouds are right at the top of boundary layer. In the western MCR, the cloud amounts are larger than that in the eastern MCR (as the air becomes moister from eastern to western MCR) and the cloud changes are also more prominent. As the dust is concentrated immediately above the low clouds, it is possible that the cloud concentrations are decreased at 900-950 hPa due to the evaporation effect. As the longwave radiation of dust is not included in the model, the heating rates changes due to LW are mainly from the changes of clouds. As cloud-top cooling and cloud-bottom heating normally occur associated with longwave (LW) radiative effects (Fu and Liou 1993), the vertical changes of clouds cause significant positive heating rates anomalies at 900-950 hPa for the western MCR (Fig. 7c) that offset the adiabatic cooling there."

Besides, the cloud changes in the western MCR are important and thus changes the temperature tendencies anomalies among different processes. In the western MCR, the

negative temperature tendency differences (Fig. S3) due to advective process are much weaker than that in the eastern MCR (Fig. S2) and thus the positive temperature anomalies are much larger than that in the eastern MCR.

[Figure]

Figure S3. Same as Fig. S2 but for the "Elevated Dust" in the western MCR.

7) Page 10, Line 10-15: What does the right panel of Figure 10 show? The dashed line vs. solid line? This should be explained in the text as well as in the figure caption.

This figure is the averaged Skew-T/ Log-P of the cycle region. The three groups of lines from left to right are dew point, temperature and the air parcel lifted respectively. It shows the vertical profile and also stability of the atmosphere of that region. As one of our coauthor is an expert on meteorology (cyclone), he suggested us to put it there for more information.

8) Page 10, Line 15-18: In Figure 11, why are there large temperature perturbations in the middle and upper troposphere? Through any mechanical pathway or numerical noises? Please clarify.

Please see the details of our concept of upward propagating of gravity wave response due to dust-induced heating in answers to question (4)

9) Page 10, Line 21-22: Figure 12c-d show the differences of temperature and RH due to associated with the dust radiative effects. How are the difference patterns linked to the case study? This is not described in the text.

We want to emphasize that the positive temperature anomaly is associated with the suppression, increased temperature lead to significant CIN increase that inhibit the convection. We also want to show another feature that the changes in circulation and weakened downward motion.

10) Page 26, Figure 10: Please state in the caption what the red dot represents.

Thanks! The red dot is the cyclone/disturbance center tracked by the GFDL vortex tracker. We have clarified in the caption.

[revised manuscript text omitted]